# Identification of PLXDC1 and PLXDC2 as the transmembrane receptors for the multifunctional factor PEDF

**Guo Cheng**[1,2†], **Ming Zhong**[1,2†], **Riki Kawaguchi**[1,2†], **Miki Kassai**[1,2], **Muayyad Al-Ubaidi**[3], **Jun Deng**[1,2], **Mariam Ter-Stepanian**[1,2], **Hui Sun**[1,2]*

[1]Department of Physiology, Howard Hughes Medical Institute, David Geffen School of Medicine, University of California, Los Angeles, Los Angeles, United States; [2]Jules Stein Eye Institute, David Geffen School of Medicine, University of California, Los Angeles, Los Angeles, United States; [3]Department of Cell Biology, University of Oklahoma Health Sciences Center, Oklahoma City, United States

**Abstract** Pigment Epithelium Derived Factor (PEDF) is a secreted factor that has broad biological activities. It was first identified as a neurotrophic factor and later as the most potent natural antiangiogenic factor, a stem cell niche factor, and an inhibitor of cancer cell growth. Numerous animal models demonstrated its therapeutic value in treating blinding diseases and diverse cancer types. A long-standing challenge is to reveal how PEDF acts on its target cells and the identities of the cell-surface receptors responsible for its activities. Here we report the identification of transmembrane proteins PLXDC1 and PLXDC2 as cell-surface receptors for PEDF. Using distinct cellular models, we demonstrate their cell type-specific receptor activities through loss of function and gain of function studies. Our experiments suggest that PEDF receptors form homooligomers under basal conditions, and PEDF dissociates the homooligomer to activate the receptors. Mutations in the intracellular domain can have profound effects on receptor activities.

*For correspondence: hsun@mednet.ucla.edu

†These authors contributed equally to this work

**Competing interests:** The authors declare that no competing interests exist.

**Reviewing editor**: Michael S Brown, The University of Texas Southwestern Medical Center, United States

## Introduction

Employing physiological pathways to impede pathological processes has been a fruitful approach in developing effective therapeutics for human disease. There exists a natural factor that can inhibit pathogenesis of several major diseases and has surprisingly diverse therapeutic value. This factor is called Pigment Epithelium-Derived Factor (PEDF) (*Dawson et al., 1999*; *Tombran-Tink and Barnstable, 2003*) and was originally identified as a strong protective factor for neurons (*Tombran-Tink and Barnstable, 2003*). It was also initially known as EPC-1, a factor that is downregulated by more than 100-fold in aged compared to young human fibroblasts (*Pignolo et al., 1993*). In an unbiased search for new antiangiogenic factors, PEDF was identified as the most potent endogenous inhibitor of angiogenesis (*Dawson et al., 1999*). PEDF inhibits endothelial cell migration and angiogenesis even in the presence of strong proangiogenic factors (*Dawson et al., 1999*). It specifically targets new vessel growth without affecting pre-existing vessels. In numerous animal models, PEDF has been shown to have potent therapeutic effects in treating several major human diseases through its neurotrophic, anti-angiogenic, antitumorigenic and antimetastatic activities. In addition to treating major blinding diseases such as ischemia-induced retinopathy, diabetic retinopathy, glaucoma and age-related macular degeneration (*Stellmach et al., 2001*; *Semkova et al., 2002*; *Miyazaki et al., 2011*), PEDF has been shown to inhibit the growth of a wide variety of cancer types including melanoma, neuroblastoma, osteosarcoma, hepatoblastoma, Lewis lung carcinoma, chondrosarcoma, gastric carcinoma, glioma, Wilm's tumor, prostate cancer, and pancreatic cancer (*Doll et al., 2003*; *Ek et al., 2006*;

**eLife digest** Many cells in our body release signals that trigger responses in other cells. A protein called PEDF is a signal released from a variety of cells that can prevent the formation of new blood vessels, protect cells in the retina and brain from damage and stop cancer cells from growing. Experiments using model animals have also demonstrated that PEDF could be used to treat a variety of eye diseases that lead to blindness and many types of cancer.

PEDF is found in tissues including the brain, eye, liver, heart and lung, but it was not known how cells sense this signal. Cells are expected to have specific proteins called receptors on the cell surface membrane to detect PEDF and transmit the signal into the cell; however, the identity of these receptors has remained a long-standing unsolved puzzle.

Cheng, Zhong, Kawaguchi et al. have now identified two human proteins that act as receptors for PEDF. These proteins—known as PLXDC1 and PLXDC2—span the cell surface membrane, and bind to PEDF on the outside of the cell. PLXDC1 and PLXDC2 are expressed on different types of cells that respond to PEDF. Furthermore, PEDF was unable to act upon cells that had been engineered to make less of these two receptors.

This study also revealed that each receptor can play different roles in different cell types. For example, exposing one type of cell from blood vessels to PEDF would normally kill them, but cells without PLXDC2 (but not those without PLXDC1) could survive PEDF treatment. Furthermore, PEDF treatment protects a type of neuron against environmental damage, and this activity depends on PLXDC1, but not PLXDC2.

How do the receptors transmit the PEDF signal from the outside of the cell to the inside of the cell? Cheng, Zhong, Kawaguchi et al. found that when PEDF is not present, both PLXDC1 and PLXDC2 form complexes containing more than one copy of either receptor. When PEDF binds to the receptors, it causes these complexes to disassemble and this activates further downstream signaling events inside the cell.

Understanding PEDF receptors and their mechanisms will open the way to developing new drugs that target these receptors to treat human diseases.

*Fernandez-Garcia et al., 2007*; *Broadhead et al., 2009*). In addition, PEDF has also been identified as a stem cell niche factor (*Pumiglia and Temple, 2006*; *Ramirez-Castillejo et al., 2006*; *Andreu-Agullo et al., 2009*; *Elahy et al., 2012*) and an anti-inflammatory factor (*Zamiri et al., 2006*; *Zhang et al., 2006b*).

PEDF is widely expressed in many tissues such as the eye, brain, spinal cord, bone, liver, heart and lung. PEDF is also naturally present in the blood (*Petersen et al., 2003*). PEDF level was found to decrease during cellular senescence and aging (*Pignolo et al., 1993*; *Tombran-Tink et al., 1995*; *Francis et al., 2004*) and in many pathological conditions. A significant decrease in PEDF level in the eyes has been observed in patients with age-related macular degeneration and diabetic retinopathy, two major blinding diseases characterized by neovascularization (*Ogata et al., 2001*; *Spranger et al., 2001*; *Holekamp et al., 2002*; *Ogata et al., 2002*; *Boehm et al., 2003*). PEDF levels were also found to decrease with age in human eyes (*Ogata et al., 2004*; *Smith and Steinle, 2007*; *Steinle et al., 2008*). PEDF expression has been inversely related to metastasis in a variety of cancer types such as gliomas (*Guan et al., 2003*), lymphangiomas (*Sidle et al., 2005*), hepatoma (*Matsumoto et al., 2004*), melanoma (*Orgaz et al., 2009*), lung cancer (*Zhang et al., 2006a*), pancreatic cancer (*Uehara et al., 2004*), and prostate cancer (*Halin et al., 2004*).

A long-standing challenge has been to understand how PEDF acts on different cell types and its fundamental transmembrane mechanisms. Uncovering the transmembrane pathways of PEDF would lead to a better understanding of its fundamental mechanisms and the development of new therapeutic strategies. After many years of effort in trying both existing and new strategies on a variety of native tissues and cell types, we were unable to identify the PEDF receptor, likely due to its low abundance and transient nature of expression. Since PEDF's actions do not match any well-characterized receptors, we reasoned that its receptor is likely new and uncharacterized. We tested human orphan receptors and transmembrane domain proteins of unknown function for their ability to bind native PEDF on the cell surface and found two transmembrane domain proteins that confer cell-surface PEDF binding

and have other properties expected of PEDF receptors. These two membrane proteins both have a large extracellular domain, a transmembrane domain, and an intracellular domain, and share about 50% homology. These two membrane proteins are called plexin domain containing 1 (PLXDC1) and plexin domain containing 2 (PLXDC2). Gene and protein expression studies have revealed overlapping but distinct tissue expression patterns of PLXDC1 (*St Croix et al., 2000*; *Gaultier et al., 2010*) and PLXDC2 (*Leighton et al., 2001*; *McMurray et al., 2008*; *Miller-Delaney et al., 2011*; *Boheler et al., 2014*).

## Results

### Cell-surface binding of PEDF mediated by PLXDC1 and PLXDC2

A prerequisite for a cell-surface receptor is the ability to confer cell-surface binding to the extracellular ligand. We found that expression of PLXDC1 or PLXDC2 confers extracellular PEDF binding to live cells (*Figure 1A–C*). PLXDC1 or PLXDC2 with a deletion of the extracellular domain no longer binds PEDF, while deletion of the intracellular domain has no effect on PEDF binding (*Figure 1D–F*). The domain structures of PLXDC1 and PLXDC2 are depicted in *Figure 1—figure supplement 1*. Staining with an extracellular epitope on live cells showed that all these proteins are expressed on the cell surface (*Figure 1G–L*). These experiments demonstrated that PEDF binds to cell-surface transmembrane domain proteins PLXDC1 and PLXDC2 through their extracellular domains.

### Cellular models of PEDF action

Studying PEDF receptors requires robust and accessible cellular assays for gain and loss of function studies. We used three cell types that respond robustly and reproducibly to PEDF as cellular models to study PEDF receptors: macrophage cell RAW267.4, endothelial cell SVEC4-10, and neuronal cell 661W. These three cell types represent three distinct cellular targets of PEDF. We found that both PLXDC1 and PLXDC2 are expressed in these three cellular models, consistent with previous knowledge that both RAW267.4 and SVEC4-10 cells express PLXDC1 (*Wang et al., 2005*; *Gaultier et al., 2010*). In macrophage RAW267.4, PEDF is known to stimulate secretion of IL-10, an anti-inflammatory cytokine (*Zamiri et al., 2006*). To perform loss of function studies, we screened for siRNAs that can effectively knockdown PLXDC1 or PLXDC2 expression in RAW26.4 cells (*Figure 2—figure supplement 1*). The most effective siRNA was used in subsequent functional assays. We found that knocking down of either PLXDC1 or PLXDC2 led to a substantial decrease in PEDF response (*Figure 2A*). Conversely, transfection of either receptor into macrophages further augments PEDF-induced secretion of IL-10 without increasing basal activity (basal activity is defined as receptor activity without PEDF treatment) (*Figure 2B*). Either receptor lacking the cytoplasmic domain no longer has this activity, consistent with role of the cytoplasmic domain in cellular signaling (*Figure 2B*). Tyrosine 481 in human PLXDC1 is a highly conserved residue in the cytoplasmic domain and is a potential phosphorylation site. PLXDC1 with a mutation of this single residue in the cytoplasmic domain (Y481F) has highly enhanced PEDF-mediated response without increasing the basal activity of the receptor (*Figure 2D,E*). One potential mechanism is that phosphorylation of this residue dampens receptor signaling and the mutation prevents this inhibition. PLXDC1 transfected cell shows about 100% increased activity in response to 2 nM PEDF as compared to control cells, while PLXDC1-Y481F cells showed about 400% increased activity in response to PEDF (*Figure 2D*). This profound stimulatory effect on PEDF signaling by mutating a single intracellular conserved residue in PLXDC1 supports its role in PEDF signaling.

Using endothelial cell SVEC4-10, we established a highly effective and reproducible assay to study PEDF-mediated endothelial cell death (*Figure 3*). We found that PEDF—mediated cell death was completely suppressed by siRNA-mediated knockdown of PLXDC2, but there was no suppression by siRNA knockdown of PLXDC1 (*Figure 3A* and *Figure 3—figure supplement 1*). Since the cytoplasmic domain of each receptor is expected to be involved in downstream signaling, we tested the effect of expression of the cytoplasmic domain fused to the transmembrane domain of another membrane protein (DCC) (*Stein and Tessier-Lavigne, 2001*) without the extracellular domain of each receptor. Interestingly, transfection of PLXDC2 cytoplasmic domain linked to the DCC transmembrane domain is sufficient to cause cell death independently of PEDF. In contrast, the cytoplasmic domain of PLXDC1 does not have this activity (*Figure 3B*). We also found that mutations in two cytoplasmic residues that are potential phosphorylation sites in the PLXDC2 tail enhance the effect of PLXDC2 in causing cell death (*Figure 3B*). These experiments demonstrated that PLXDC2 is responsible for mediating

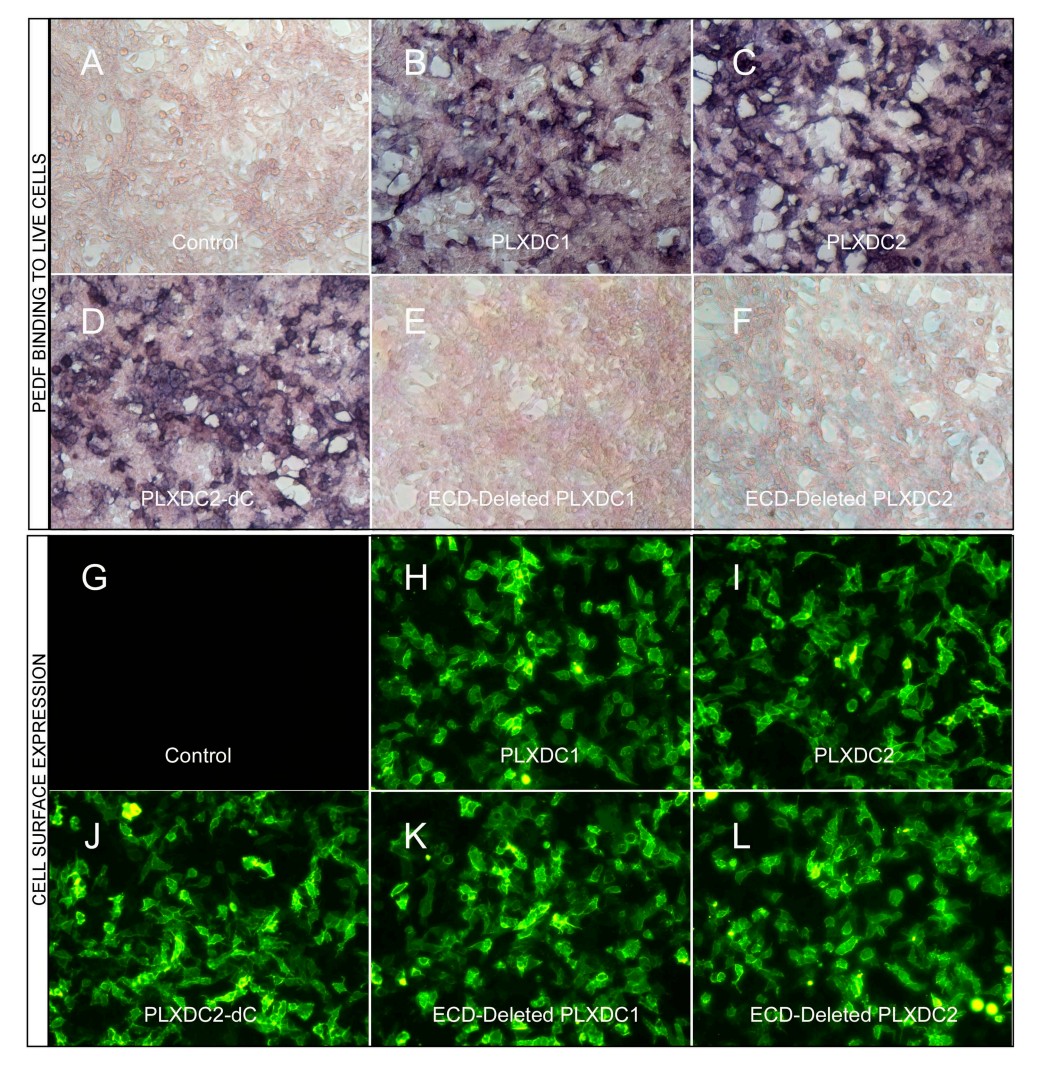

**Figure 1**. The binding of PEDF to PLXDC1 and PLXDC2 on cell surface. Upper panel: Binding of biotinylated PEDF to control HEK293 cells (**A**) or HEK293 cells transfected with PLXDC1 (**B**), PLXDC2 (**C**), intracellular domain deleted PLXDC2 (**D**, PLXDC2-dC), extracellular domain (ECD) deleted PLXDC1 (**E**), or ECD deleted PLXDC2 (**F**). Binding was detected by stretavidin-alkaline phosphatase, shown as deep purple color. Lower panel: Live cell staining of an epitope tag of control HEK293 cells (**G**) or HEK293 cells transfected with PLXDC1 (**H**), PLXDC2 (**I**), intracellular domain deleted PLXDC2 (**J**, PLXDC2-dC), ECD-deleted PLXDC1 (**K**), or ECD deleted PLXDC2 (**L**). All constructs have the epitope tag engineered after the secretion signal at the N-terminus.
The following figure supplement is available for figure 1:

**Figure supplement 1**. PLXDC1 and PLXDC2 schematic diagrams and alignment showing the definitions of domains.

PEDF-mediated cell death in this cell type, while there is no detectable role of PLXDC1. In addition, the cytoplasmic domain of the PLXDC2 is sufficient to trigger downstream activity.

To assay PEDF's neurotrophic activity, we used 661W cells, a neuronal cell line derived from cone photoreceptors (*Tan et al., 2004*; *Kanan et al., 2008*). We found that PEDF treatment effectively protects 661W cells against oxidative damage. Knocking down PLXDC1, but not PLXDC2, in 661W cells abolishes the protective effect of PEDF (*Figure 4A* and *Figure 4—figure supplement 1*). Conversely, using gain of function analysis, we showed that transfection of PLXDC1 further enhances the protective effect of PEDF and that transfection of the cytoplasmic domain of PLXDC1 protects 661W

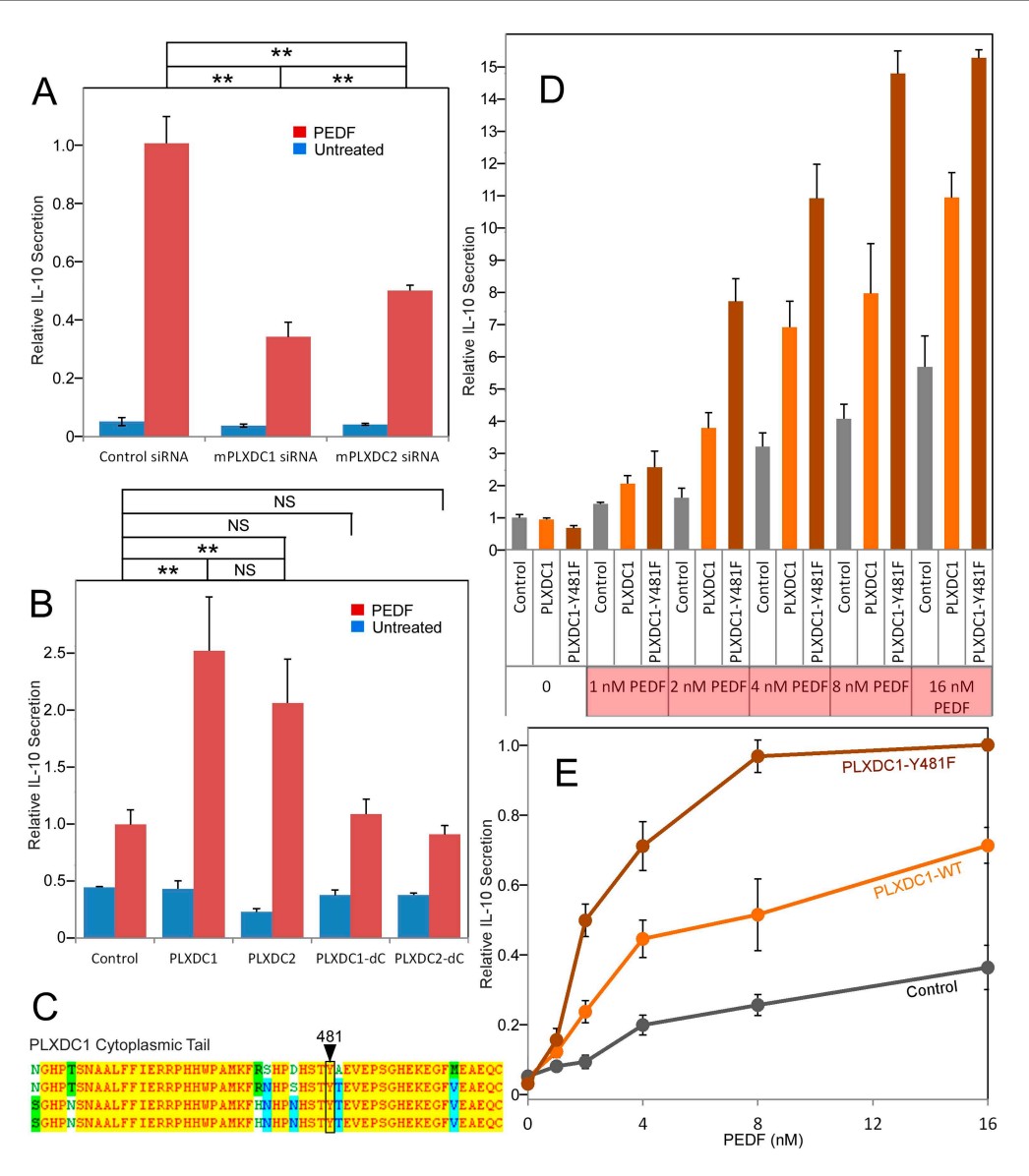

**Figure 2**. The roles of PLXDC1 and PLXDC2 in PEDF-induced IL-10 secretion by macrophage cell RAW267.4.
(**A**) siRNA-mediated knockdown of PLXDC1 or PLXDC2 substantially suppresses PEDF-stimulated IL-10 secretion.
Activity of control transfected cells with PEDF treatment is defined as 1. ** = p < 0.01. (**B**) Transfection of either
PLXDC1 or PLXDC2 cDNA enhances RAW cell's PEDF-stimulated IL-10 secretion, while PLXDC1 or PLXDC2 lacking
the cytoplasmic domain (PLXDC1-dC or PLXDC2-dC) do not show significantly different secretion from control
EGFP transfection. Activity of control cells with PEDF treatment is defined as 1. ** = p < 0.01; NS = not significant.
(**C**) Alignment of human, bovine, mouse, and rat PLXDC1 cytoplasmic tail and the location of the putative
phosphorylated residue (residue number according to human PLXDC1). (**D**) Comparing PEDF-induced IL-10
secretion by RAW267.4 transfected with PLXDC1 and PLXDC1-Y481F. Mutation Y481F on the cytoplasmic tail
of PLXDC1 greatly enhances its response to PEDF. Activity of control transfected cells without PEDF treatment
is defined as 1. (**E**) PEDF concentration-dependent stimulation of IL-10 secretion from control, PLXDC1, and
PLXDC1-Y481F transfected cells (from **D**). Activity of PLXDC1-Y481F cells at 16 nM PEDF is defined as 1.

The following figure supplements is available for figure 2:

**Figure supplement 1**. Unbiased screening for effective siRNAs that knock down PLXDC1 or PLXDC2 expression in
macrophage cell RAW267.4.

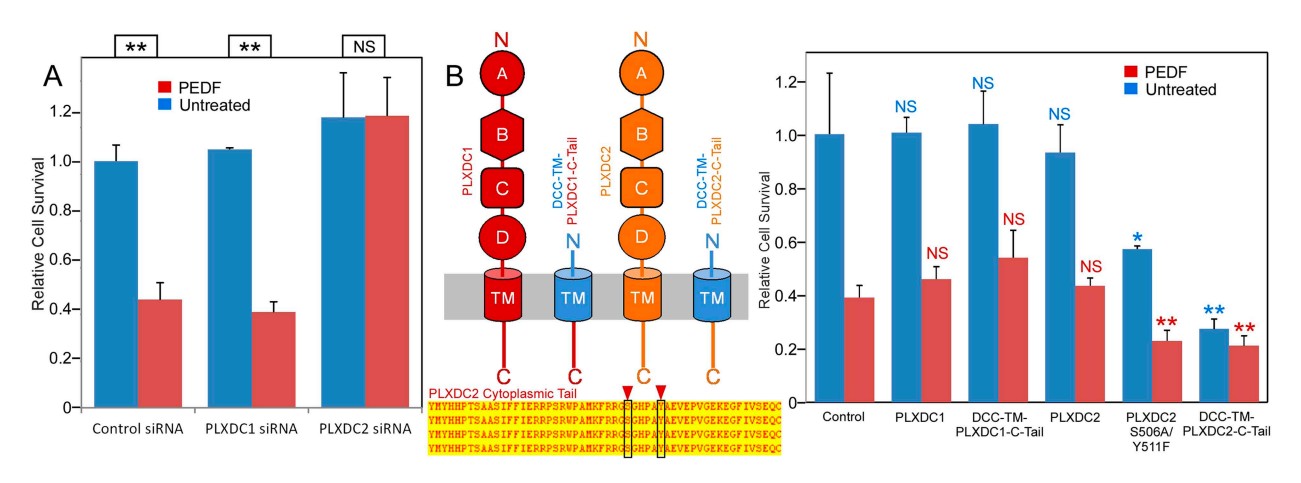

**Figure 3**. The PLXDC2 dependence of PEDF's effect on endothelial cell SVEC4-10. (**A**) PEDF promoted-cell death of SVEC4-10 cells is suppressed by siRNA-mediated knockdown of PLXDC2, but not PLXDC1. Survival of control siRNA tranfected cells without PEDF treatment is defined as 1. Statistical significance is shown on the top. ** = p < 0.01, and NS = not significant. (**B**) Left panel: Schematic diagrams of full length receptors and the fusion proteins for the receptor cytoplasmic tails. The cytoplasmic tail of the receptor is fused to the TM domain of DCC, which is fused to the secretion signal of alkaline phosphase at the N-terminus. Alignment of human, mouse, rat and bovine PLXDC2 cytoplasmic tails shows complete conservation (bottom). Locations of potential phosphorylation sites are indicated. Right panel: Expression of cytoplasmic tail of PLXDC2, but not the cytoplasmic tail of PLXDC1 promotes SVEC4-10 cell death. PLXDC2 double mutant S506A/Y511F has greater activity. Survival of control EGFP transfected cells without PEDF treatment is defined as 1. Statistical significance of the comparison of cells without PEDF treatment (with the control cells without PEDF treatment) is shown in blue. Statistical significance of the comparison of cells with PEDF treatment (with the control cells with PEDF treatment) is shown in red. * = p < 0.05, ** = p < 0.01, and NS = not significant.

The following figure supplement is available for figure 3:

**Figure supplement 1**. Unbiased screening for effective siRNAs that knock down PLXDC1 or PLXDC2 expression in endothelial cell SVEC4-10.

independent of PEDF (*Figure 4B*). These experiments suggest that the cytoplasmic domain of the receptor is responsible for triggering the intracellular events during receptor activation. The dependence of 661W cell on PLXDC1 for the survival promoting effect of PEDF is in contrast to SVEC4-10's dependence on PLXDC2 to mediate the cell death effect of PEDF.

## PEDF's receptor activation mechanism

Although PLXDC1 and PLXDC2 have an architecture reminiscent of membrane receptors with extracellular, transmembrane and intracellular domains, they do not belong to any well-characterized receptor families and represent a family of their own. What is the consequence of PEDF's binding to these receptors? To answer this question, we performed further mechanistic studies on PEDF/receptor interaction. We hypothesized that PEDF might affect the oligomerization states of the receptors. We found that PLXDC1 and PLXDC2 form homooligomers in the absence of PEDF. We designed experiments to compare homooligomerization and heterooligomerization and found that PLXDC1 and PLXDC2 preferentially form homooligomers (*Figure 5—figure supplement 1*). Using deletion series of the extracellular domains, we identified domain D as an important domain for oligomerization (*Figure 5A*). We noticed that the last residue in the cytoplasmic domain of both PLXDC1 and PLXDC2 is a conserved cysteine. Since intracellular cysteines are in the reduced state, we tested whether copper phenanthroline-induced thiol oxidation (*Zhou et al., 2009*) could crosslink these cysteines. Indeed, we found that copper phenanthroline promoted the formation of covalent dimers, consistent with the close proximity of the two cysteines on the cytoplasmic domain (*Figure 5B*). This covalent dimer can be cleaved under reducing conditions (*Figure 5B*), consistent with its linkage through a disulfide bond catalyzed by copper phenanthroline. Interestingly, incubation with PEDF before copper phenanthroline treatment inhibited the formation of the dimer and promoted the formation of the monomer, indicating that the cytoplasmic tail is no longer in close contact with another cytoplasmic tail (*Figure 5B*).

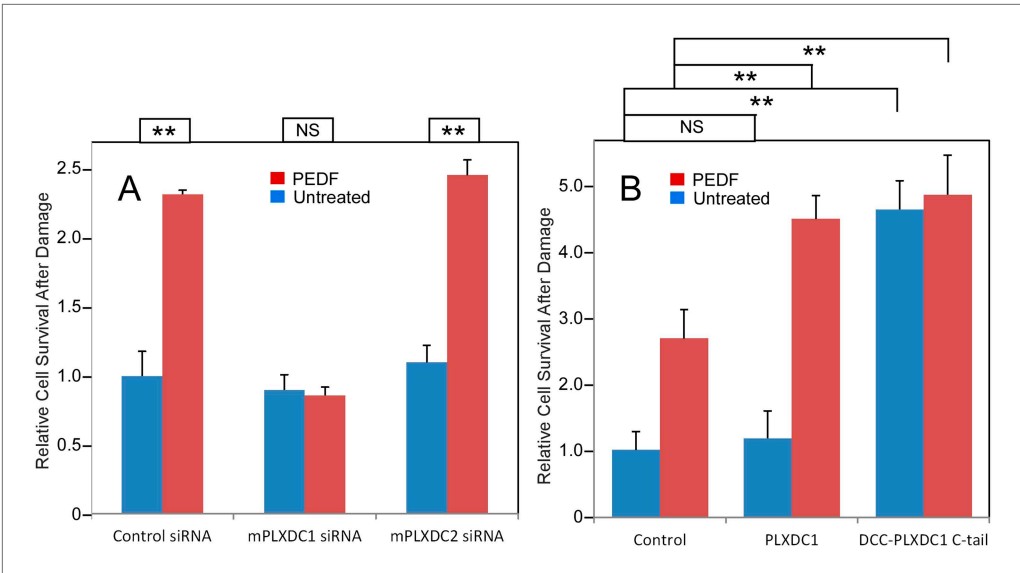

**Figure 4**. PEDF's neurotrophic effect on 661W cells depends on PLXDC1. (**A**) PEDF treatment protects 661W cells against hydrogen peroxide-mediated oxidative damage. siRNA-mediated knockdown of PLXDC1, but not PLXDC2 abolishes the protective effect of PEDF. The survival of control siRNA transfected cell without treatment is defined as 1. (**B**) Transfection of PLXDC1 or the cytoplasmic domain of PLXDC1 fused to DCC's TM domain enhances protection of 661W cells against damage caused by hydrogen peroxide. The effect of DCC-PLXDC1 C-tail is independent of PEDF. Cell survival of transfected control is defined as 1. Statistical significance is shown on the top. ** = p < 0.01; NS = not significant.

The following figure supplement is available for figure 4:

**Figure supplement 1**. Unbiased screening for effective siRNAs that knock down PLXDC1 or PLXDC2 expression in neuronal cell 661W.

---

To pinpoint the receptor domain that interacts with PEDF, we used a deletion series of PLXDC1 to perform copurification analysis with PEDF and found that extracellular domain B plays an important role in binding to PEDF (*Figure 6A*). To demonstrate PEDF's effect on receptor oligomerization in live cells, we developed an assay to visualize PEDF-induced receptor dissociation on the cell surface. Epitope-tagged extracellular domain of PLXDC1 is associated with the cell surface through its binding to the coexpressed full length untagged PLXDC1 (*Figure 6B*). Incubation of the cells with PEDF causes the dissociation of the extracellular domain and the loss of the epitope tag from the cell surface (*Figure 6B*). This live cell-based assay again demonstrated the ability of PEDF to dissociate receptor oligomer.

To further observe PEDF's effect on receptor oligomerization in real time, we developed a fluorescence resonance energy transfer (FRET)-based assay. We coexpressed PLXDC1 fused to a Cyan Fluorescent Protein (CFP) and PLXDC1 fused to a yellow fluorescent protein (YFP) at the C-terminus and observed a time-dependent decrease in FRET signals after addition of PEDF, but not a control extracellular protein nidogen (*Figure 6C*). To show that this decrease in FRET signal was due to receptor dissociation, we crosslinked the cysteine residues on the cytoplasmic domain of PLXDC1 using sulfhydryl-specific crosslinker bismaleimidoethane (BMOE) before PEDF addition and found that this crosslinker prevents PEDF-dependent suppression of the FRET signal (*Figure 7*). However, mutation of the cysteine to serine (C500S) prevents the blocking effect of BMOE on PEDF (*Figure 7*).

In summary, using three different techniques (visualization of receptor oligomers in SDS-PAGE, visualization of receptor oligomers on the cell surface, and tracking receptor interaction in real-time), we found that PEDF receptors self-associate to form homooligomers and that PEDF has the ability to dissociate receptor homooligomers so that the cytoplasmic tails are no longer in contact with each other. This effect of PEDF is consistent with the ability of membrane-tethered cytoplasmic tail to activate downstream pathways.

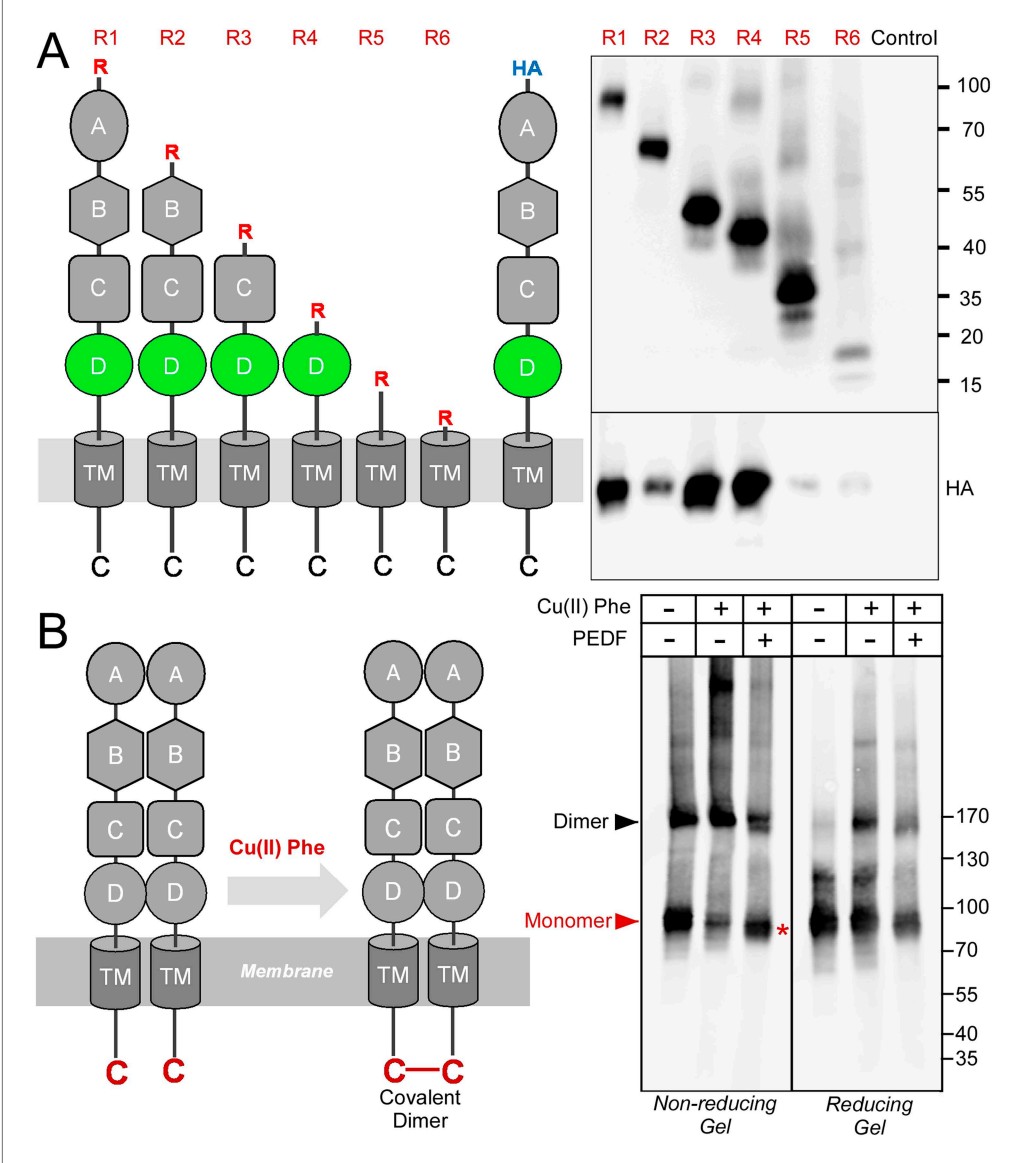

**Figure 5**. Receptor oligomerization. (**A**) PLXDC1 deletion series with a Rim tag following the N-terminal secretion signal were purified together with HA-tagged full length PLXDC1 by purifying the Rim tag (diagrams on the left). Anti-Rim Western is shown on the top and anti-HA Western is shown on the bottom for the elutions. HA-PLXDC1 no longer copurifies if domain D is deleted. (**B**) Copper phenanthroline [Cu (II) Phe] treatment creates covalent receptor dimer through oxidation of the free cysteine residue on the cytoplasmic tail (schematic diagram on the left). Cu (II) Phe oxidation creates disulfide bond-linked covalent PLXDC1 dimer, as indicated in the Western blot for the receptor. PEDF inhibits dimer formation as shown by increased monomer band on a non-reducing gel after Cu (II) Phe oxidation (red asterisk). The disulfide bond-linked dimers are sensitive to DTT treatment as shown in the reducing gel on the right. Molecular weight markers are in kD.

The following figure supplement is available for figure 5:

**Figure supplement 1**. PLXDC1 and PLXDC2 preferentially form homooligomers.

## Discussion

Both PLXDC1 and PLXDC2 are cell-surface transmembrane domain proteins and confer cell-surface binding to PEDF, as expected of cell-surface receptors. PLXDC1 and PLXDC2 represent a receptor family of their own that has two members in human. Through both loss of function and gain of

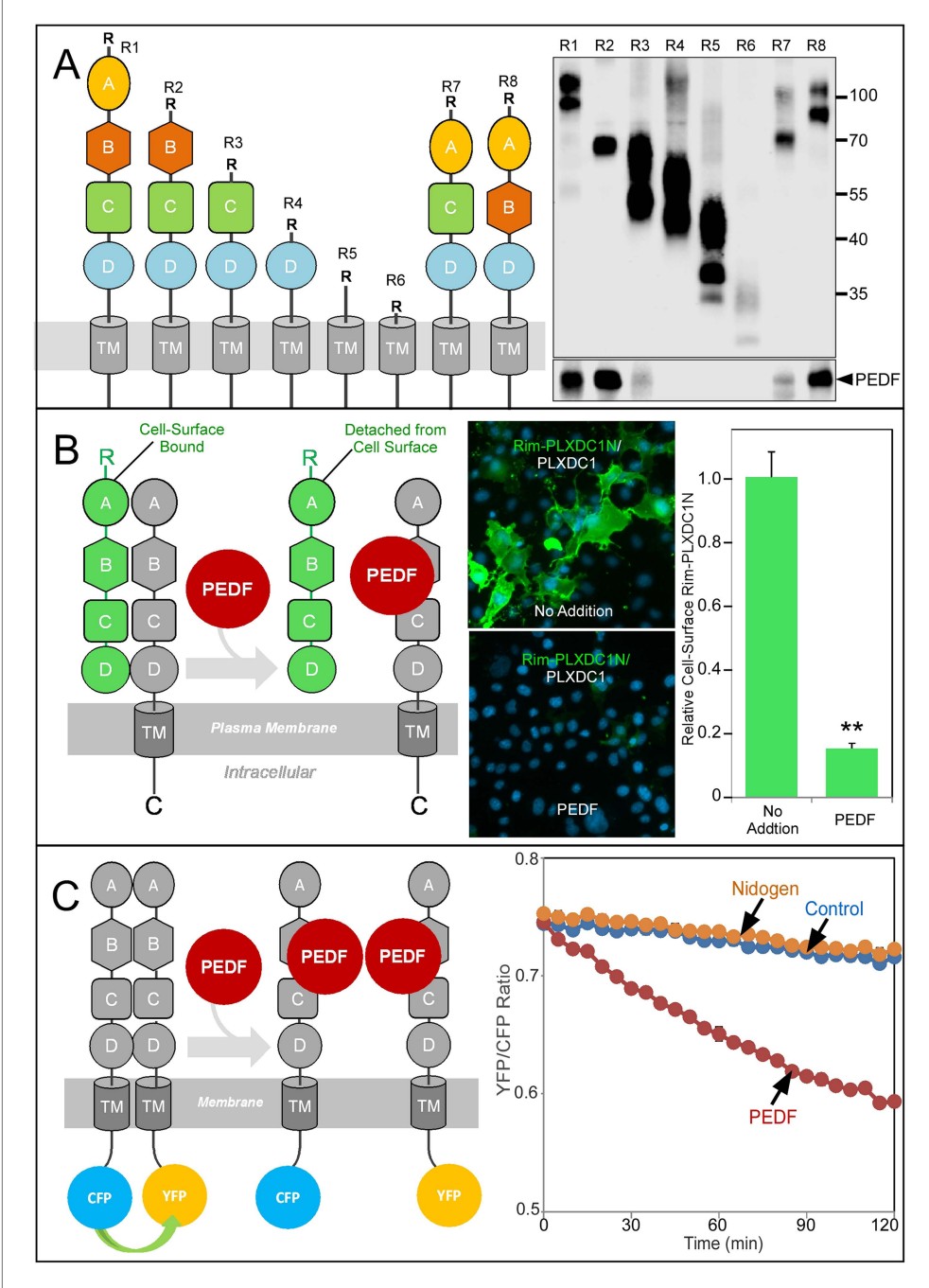

**Figure 6**. Structure/function and real-time analysis of PEDF/receptor interaction. (**A**) PEDF with HA tag after the N-terminal secretion signal is copurified with different deletion mutants of PLXDC1, which are all tagged with a Rim tag following the N-terminal secretion signal (diagrams on the left). Purification of the Rim-tagged proteins with HA-PEDF is shown on the right. The upper Western is the anti-Rim Western and the lower Western is anti-HA Western to detect PEDF. Deletion of domain B largely abolishes the interaction between PLXDC1 and HA-PEDF. (**B**) An assay to study PEDF-mediated disruption of receptor dimers on the cell surface. Schematic diagram is on the left. PEDF displaces Rim-tagged PLXDC1 extracellular domain (green) bound to cell surface PLXDC1 (gray) to cause detachment of the extracellular domain. Immunostaining of Rim-tagged PLXDC1 extracellular domain, Rim-PLXDC1N, (cotransfected with PLXDC1) on live cell surface is shown in the middle two pictures (green signal). Blue color is nucleic acid stain DAPI. Upper picture: control (no PEDF). Lower picture: PEDF treated. Quantitation of bound Rim-tagged PLXDC1 extracellular domain on the cell surface with or without PEDF treatment is shown on

*Figure 6. Continued on next page*

*Figure 6. Continued*

the right. (**C**) An assay to study PEDF-mediated disruption of receptor dimerization in real time. Schematic diagram of the experimental design is shown on the left. The cytoplasmic tail of PLXDC1 is linked to CFP or YFP, which are in close proximity due to receptor dimerization. PEDF suppresses the FRET signal between CFP and YFP if it disrupts the association of the receptor dimer. Right panel: PEDF, but not a control protein (nidogen) causes a time-dependent decrease in FRET signal between PLXDC1-CFP and PLXDC1-YFP. Both PEDF and nidogen were added at time 0.

function studies using three distinct cellular models that respond robustly to PEDF, we showed that both receptors have the expected properties of PEDF receptors in transducing PEDF signal and have cell type-specific roles. How does PEDF activate its cell-surface receptors? Our experiments suggest that these receptors form homodimers under basal conditions, and the dimerization functions to inhibit self-activation. PEDF activates its receptors through dissociation of the dimer. PEDF's ability to dissociate the receptor dimer was demonstrated using three independent techniques. Without oligomerization provided by the extracellular domains, the cytoplasmic domain of the PEDF receptor is sufficient to activate downstream signaling. We also showed that modulating the cytoplasmic domain strongly affects receptor signaling. For example, a mutation in a conserved tyrosine residue in the cytoplasmic domain of PLXDC1 greatly enhanced PEDF-induced and PLXDC1-dependent IL-10 secretion from macrophages. Structure and function analysis revealed an important PEDF-interacting domain and a dimerization domain in the receptors.

PLXDC1 and PLXDC2 are homologous membrane proteins with overlapping but distinct expression patterns in physiological and pathological conditions, as revealed by gene and protein expression studies. PLXDC1 (also called tumor endothelial marker 7) was discovered as one of the genes enriched in many types of human tumor endothelial cells (*St Croix et al., 2000*; *Schwarze et al., 2005*; *Beaty et al., 2007*; *Lu et al., 2007*; *van Beijnum et al., 2009*). PLXDC1 was also found to be highly expressed in the endothelial cells of another human disease- diabetic retinopathy- and is highly specific to diseased blood vessels (*Yamaji et al., 2008*). PLXDC1 is also expressed on the macrophage cell surface (*Gaultier et al., 2010*) and is downregulated by LRP-1, a large membrane protein involved in endocytosis of a variety of cell surface receptors (*Herz et al., 1990*; *Herz and Strickland, 2001*). In contrast, PLXDC2, but not PLXDC1, was found as one of the genes that increase in expression during cellular senescence (*Schwarze et al., 2005*), as one of the E2F1 target genes repressed by serum (*Hallstrom et al., 2008*), as a gene negatively correlated with malignant cell transformation in tumors and its disturbance increases tumor volume (*McMurray et al., 2008*), and as a candidate axon guidance molecule (*Leighton et al., 2001*). Microarray analysis also revealed PLXDC2, but not PLXDC1, as one of the markers of adult stem cells (*Noh, 2006*). Proteomic analysis also revealed PLXDC2 on the cell surface of human pluripotent stem cells (*Boheler et al., 2014*). PLXDC2 is also known as a mitogen for neuroprogenitors (*Miller-Delaney et al., 2011*). Interestingly, expression patterns of both PEDF and PLXDC2 have been previously linked to cellular growth states. PEDF expression correlates with G0 growth arrest in fibroblasts (*Pignolo et al., 1993*, *2003*) and has been demonstrated to induce cell cycle arrest of glioma cells (*Zhang et al., 2007*). The correlation between PLXDC2 and cell proliferation (*Miller-Delaney et al., 2011*), senescence (*Schwarze et al., 2005*), transformation (*McMurray et al., 2008*) and cell death (*Hallstrom et al., 2008*) has been noted in several studies. Consistent with these earlier reports, our study identified PLXDC2 as playing the dominant role in a PEDF-induced endothelial cell death model.

From a different perspective, this study identified the extracellular ligand for two membrane proteins and showed that they function as cell-surface receptors that can transduce an extracellular signal. These receptors represent a new type of cell-surface receptor. They oligomerize in the basal state and are activated by ligand-induced dissociation. Mechanistically, they behave oppositely from many known single transmembrane domain signaling receptors that are activated by ligand-induced dimerization. However, detailed mechanisms can be complex, as exemplified by the human grown hormone receptor. For a long time after its original discovery more than two decades ago (*Leung et al., 1987*; *Cunningham and Wells, 1989*; *Cunningham et al., 1989*, *1991a*, *1991b*), the growth hormone receptor was hypothesized to be activated by ligand-induced receptor dimerization, as revealed by classic studies. A recent study suggested that its conformational change is similar to that of scissors (*Brooks et al., 2014*).

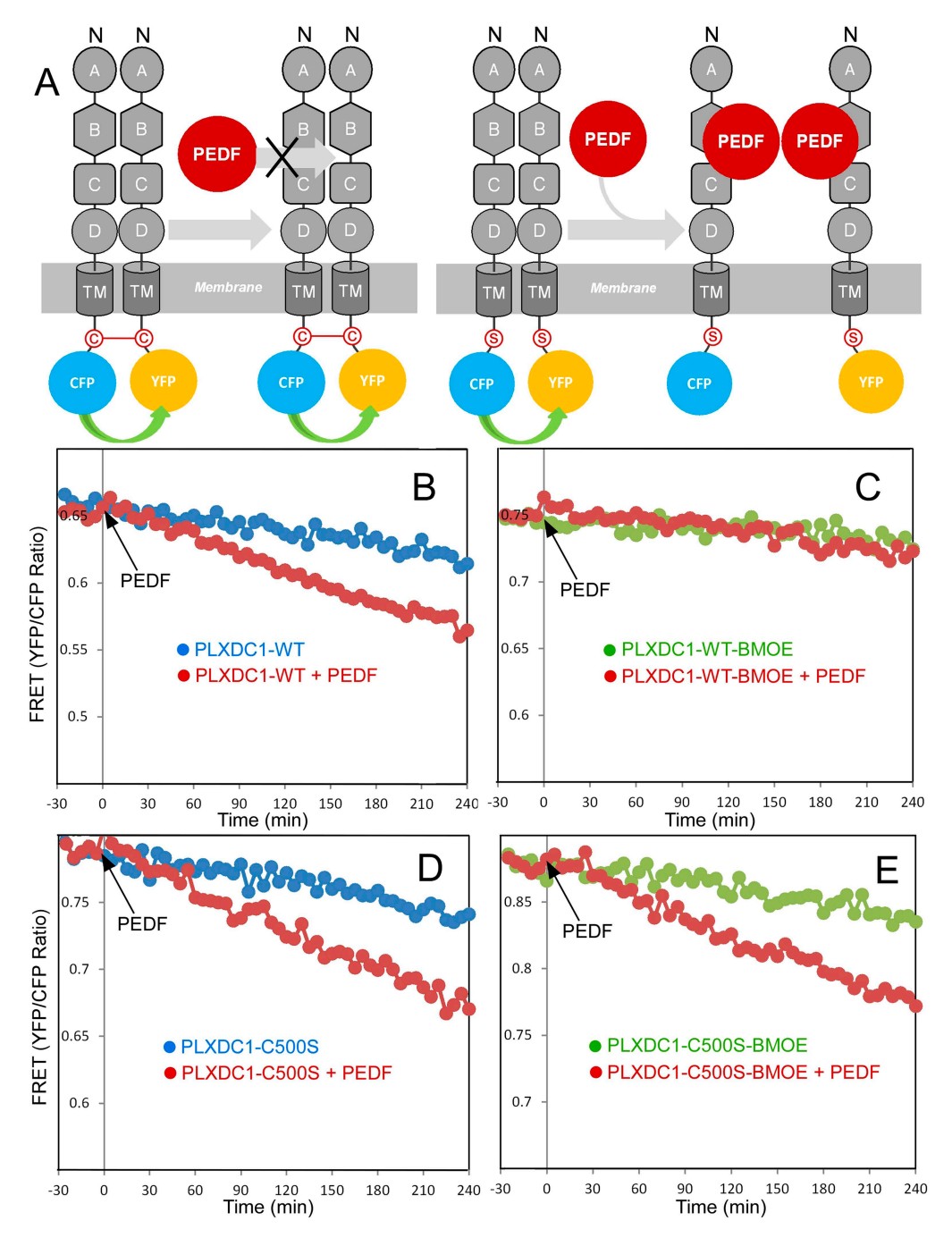

**Figure 7**. Crosslinking of the terminal cysteine of PLXDC1 prevents PEDF's effect on receptor dimerization. (**A**) Schematic diagrams of the experimental design. 'C' indicates the cysteine residue located at the C-terminus of PLXDC1. 'S' indicates mutation of this residue to serine. (**B** and **C**) Crosslinking of the cysteine on the cytoplasmic tail of wild-type PLXDC1 (PLXDC1-WT) by sulfhydryl-specific crosslinker BMOE prevents the PEDF-dependent decrease of FRET signal between PLXDC1-CFP and PLXDC1-YFP. (**D** and **E**) BMOE has no effect on PEDF-dependent decrease of FRET signal between PLXDC1-CFP and PLXDC1-YFP if the C-terminal cysteine is mutated to serine (PLXDC1-C500S). DOI: 10.7554/eLife.05401.014

PEDF receptors have at least two mechanisms to diversify their function and regulation. First, both human PLXDC1 and PLXDC2 have several isoforms. Second, receptors can couple to different immediate downstream molecules. The immediate downstream molecules that transduce the PEDF

receptor signal (the equivalent of G-proteins for G-protein coupled receptors) are still unknown. As shown by G-protein coupled receptors, distinct cellular responses can be governed not only by distinct cell-surface receptors, but also by distinct downstream molecules. Because PEDF is a multifunctional factor, understanding the fundamental mechanisms of its transmembrane receptors in different cellular contexts will help to develop potent and specific small molecule-based therapeutics in treating diseases.

## Materials and methods

### PEDF purification from conditioned media

To purify PEDF, we performed large-scale transfection of human PEDF cDNA into HEK293T cells using Jetprime reagent (Polyplus-transfection SA, Illkirch, France). Six hours after transfection, the cells were washed twice with Hank's Balanced Salt Solution (HBSS) and changed to serum free medium (SFM). PEDF is naturally present in several isoforms as revealed by isoelectric focusing (*Tombran-Tink et al., 1995*). Not all PEDF isoforms are equally active (*Duh et al., 2002*; *Subramanian et al., 2012*). Consistent with previous reports (*Duh et al., 2002*; *Subramanian et al., 2012*), we found that more negatively charged PEDF isoforms are more biologically active. Therefore, we purified PEDF using sequential anion exchange chromatography. Briefly, PEDF was purified from conditioned SFM 24, 48, or 72 hours after transfection using a combination of Q sepharose (Amersham/GE Healthcare, Little Chalfont, United Kingdom) and polyethyleneimine column (AX-300, Eprogen, Downers Grove, IL). The conditioned SFM was dialyzed against binding buffer (20 mM Tris, pH 7.5 and 50 mM NaCl) overnight at 4°C. Dialyzed medium was then applied to Q sepharose equilibrated with the binding buffer. The column was washed with 10 bed volumes of binding buffer before elution using buffer containing 20 mM Tris, pH 7.5 with 100, 200, 300, or 400 mM NaCl. High performance liquid chromatography (HPLC) using polyethyleneimine column was performed to further purify PEDF. Briefly, buffer exchange for the Q fraction containing PEDF was performed for three times by diluting with 10 vol of column buffer (25 mM Tris, pH 8.4) and concentrating in an Amicon Ultra-4 filter (Millipore, Billerica, MA) before being loaded onto the HPLC system. HPLC was performed using the Agilent 1100 series liquid chromatography system with a diode-array detector. Proteins were separated on polyethyleneimine column using column buffer with increasing NaCl concentrations as the mobile phase (from 0 to 1.8 M in 8 min and the 1.8 M sodium concentration was maintained for another 4 min). The flow speed of mobile phase was 0.5 ml/min, and four fractions were collected every minute. Elution fractions with significant A280 value were saved. Buffer exchange was performed for three times by diluting with 10 vol of phosphate buffered saline (PBS) and concentrating in an Amicon Ultra spin filter. The final volume of each concentrated elution was 0.5 ml. The presence of PEDF was confirmed by both total gel staining and Western blot analysis. Protein sterilization was achieved using Ultrafree Durapore 0.22 µm filter (Millipore). We also produced PEDF with a 6XHis tag followed by the HA tag at the N-terminus after the secretion signal and PEDF with an 8XHis tag at the C-terminus of PEDF. We found that tagging significantly diminishes the biological activity of PEDF. Therefore, all biological assays were performed using untagged PEDF. The negative effect of epitope tagging on PEDF is likely one of the reasons that PEDF receptors are difficult to identify.

### Engineering cDNAs for PLXDC1 and PLXDC2

The domains for human PLXDC1 (numbered according to the full length receptor with the secretion signal) were: domain A (19–127), domain B (128–242), domain C (243–292), and domain D (293–359). We also created chimeras containing the transmembrane domain (TM) of another single transmembrane protein DCC (*Stein and Tessier-Lavigne, 2001*) and the C-terminus of the receptors. The DCC TM domain was fused to the secretion signal of alkaline phosphatase and Rim tag at the N-terminus and the cytoplasmic tail of PLXDC1 or PLXDC2 at the C-terminus. A monoclonal antibody has been produced against the Rim tag, which has 14 residues (NETYDLPLHPRTAG) (*Illing et al., 1997*). The most likely positions of phosphorylation sites in the cytoplasmic domains of human PLXDC1 and PLXDC2 were identified through PhosphoSite Plus, a bioinformatics resource to identify potential protein phosphorylation sites (http://www.phosphosite.org/staticUsingPhosphosite.do). Residue numbers are according to human isoform 1. The secretion signal for alkaline phosphatase (AP) followed by the Rim tag was engineered at the N-terminus of each extracellular domain.

### Receptor binding assay

PEDF was biotinylated using sulfo-NHS-SS-biotin (Pierce, Rockford, IL) after overnight dialysis in PBS at 4°C. After biotinylation, free biotin was removed by further overnight dialysis in PBS at 4°C and the

degree of biotinylation was assessed by visualizing the shifting of molecular weight in SDS-PAGE gels after incubation with streptavidin. The advantage of biotinylation is that biotin is a tag much smaller than peptide tags or fusion protein tags and is less likely to interfere with biological activities. In addition, biotin is added after protein production and folding and allows sensitive detection. However, excessive biotinylation can inactivate proteins due to the modification of key lysine residues. To prevent excessive biotinylation, the ideal degree of biotinylation is about 90%, as judged by shifting in molecular weight after binding to streptavidin. Biotinylated PEDF (20 nM) was added to transfected or control cells grown on a fibronectin-coated dish in HBSS with 10 mM HEPES, pH 7.5 and 2 mg/ml BSA at room temperature for 1 hour. After two continuous washes with HBSS, 10 mM HEPES, pH 7.5, the cells were fixed using freshly made 4% paraformaldehyde in HBSS, pH 7.5 for 20 min. The cells were heated in HBSS at 65°C for 1 hr to inactive endogenous AP activity. After blocking in 5 mg/ml BSA in PBS for 1 hour, the cells were incubated with streptavidin-AP diluted in 5 mg/ml BSA in PBS. After four washes using PBS, AP activity was visualized using NBT/BCIP (Thermo Scientific, Waltham, MA).

## Receptor dimer crosslinking catalyzed by copper phenanthroline [Cu (II) Phe]

Membranes were prepared from HEK293 cells transfected with the receptors using PBS and 5 mM EDTA, which helps to keep free cysteine residues in the reduced state. After one wash using PBS, the membrane was resuspended in PBS and incubated with or without PEDF for 3 hr at room temperature. Oxidation-induced disulfide bond formation was catalyzed by 0.5 mM Cu (II) Phe. After 5 min, EDTA was added to each reaction to 50 mM to stop the oxidation reaction. The membranes were spun down, resuspended and boiled in SDS loading buffer with or without DTT for loading onto a SDS-PAGE gel.

## Assay to visualize the dissociation of receptor extracellular domain by PEDF

We developed a live cell-based assay to study the ability of PEDF to dissociate receptor complexes. PLXDC1 extracellular domain with a Rim tag following the N-terminal secretion signal of alkaline phosphatase is cotransfected into COS-1 cells with wild-type PLXDC1 with no tag. The Rim-tagged extracellular domain of PLXDC1 associates with the cell surface through its interaction with the extracellular domain of the full length PLXDC1. 1 day after transfection, the cells were washed once with HBSS and incubated overnight at 37°C in SFM with 5 mg/ml BSA with or without 50 nM PEDF. The next day cell surface associated Rim tagged protein is assessed through live cell staining by anti-Rim antibody by incubating with antibody diluted in SFM with 5 mg/ml BSA for 60 min at 37°C. After antibody binding, the cells were washed with HBSS and fixed in freshly made 4% paraformaldehyde in HBSS, pH 7.5 for 20 min. Rim antibody was detected through immunostaining using anti-mouse secondary antibody. Fluorescent signals were quantified using Nikon NIS Elements AR Analysis software.

## Cell death assay

The survival of SVEC4-10 endothelial cells (ATCC, Manassas, VA) upon PEDF treatment was analyzed by the MTT assay. Briefly, SVEC4-10 cells were grown in 10% FBS in DMEM containing penicillin and streptomycin until confluency. Cell death was initiated by splitting confluent SVEC 1:10 in serum free media (SFM) and 1 mg/ml BSA with or without 20 nM PEDF. PEDF was added before cell addition. Trypsin used in cell splitting was neutralized by defined trypsin inhibitor (Gibco) and removed by spinning down the cells. Cell viability was assessed 24 hr later after cell plating. MTT assay was done by incubating cells with 100 µg/ml MTT reagent (3-(4,5-dimethylthiazol-2-yl)-2,5-diphenyltetrazolium bromide) in SFM for 3 hr at 37°C. Dimethyl sulfoxide (DMSO) was added to each well after MTT was removed. The absorbance of the purple color from the formazan formed was measured and quantified using POLARstar Omega (BMG Labtech, Ortenberg, Germany) at 534 nm. For cell death assay on siRNA transfected cells, cell plating in SFM (with or without PEDF addition) was done after two rounds of 48-hour transfection. For cell death assay on DNA transfected cells, the cells were split at 1:15 ratio 16 hours before transfection and were transfected using Jetprime reagent (about 40–50% confluency during transfection). Cell plating in SFM and 1 mg/ml BSA (with or without PEDF addition) was done 16 hours after transfection. All assays were performed in 96-well plates in triplicate.

## Assay for neurotrophic activity of PEDF

We found that PEDF protects cone-derived 661W cells from hydrogen peroxide-mediated oxidative damage. 661W cells were grown in 10% FBS, 40 µg/l of hydrocortisone 21-hemisuccinate, 40 µg/l of

progesterone, 32 mg/l of putrescine, 40 µl/l of β-mercaptoethanol in DMEM containing penicillin and streptomycin and treated with 10 nM PEDF for 20 hours. After addition of 2.5 mM $H_2O_2$ for 1 hr (to achieve about 90% cell death in control cells the next day), the media was replaced with fresh media and the cells were continuously grown for 24 hours. Cell survival was quantified using the MTT assay as described above. For siRNA transfected cells, 10 nM PEDF was added after two rounds of 48-hour transfection. For DNA transfection, the cells were split at 1:5 ratio 16 hours before transfection and were transfected using Jetprime reagent. Four hours after transfection, the media was changed to fresh media and PEDF was added.

### Assay for IL-10 secretion by macrophage

Macrophage cell line RAW267.4 (ATCC) was grown in RPMI1640 media with 10% FBS (Gibco Certified Performance Plus FBS), penicillin and streptomycin. For DNA transfection, the cells were split at 1:6 ratio 16 hours before transfection and were transfected using Jetprime reagent. Eight hours after transfection, the media was changed to fresh media and PEDF was added. PEDF-induced IL-10 secretion from macrophages was assayed using mouse IL-10 ELISA kit from Southern Biotech 18 hours after PEDF addition. For siRNA transfection, media was changed to fresh media and PEDF was added after two rounds of 48-hour transfection. All final assays were performed in 96-well plates in triplicate. We found that longer culture of RAW267.4 cells without cell replating leads to more responsiveness to PEDF. Since siRNA experiments need longer cell culture time, cells are consistently more responsive to PEDF than cells in DNA transfection experiments, which are done only 1 day after transfection.

### siRNA-mediated knockdown

After screening many siRNA transfection reagents including X-tremeGENE (Roche, Basel, Switzerland), siTran (Origene, Rockville, MD), Jetprime (Polyplus-transfection SA), RNAiMAX (Life Technologies, Carlsbad, CA), and GenMute (SignaGen, Rockville, MD), we found that the most effective siRNA transfection reagent is RNAiMAX. Since the three cell types (RAW267.4, SVEC4-10 and 661W) that were used as cellular models to study PEDF receptors are all mouse cells, siRNAs targeting mouse genes were tested. For mouse PLXDC1, siRNAs tested included Dharmacon D-060224-01 (siRNA-1), Dharmacon D-060224-02 (siRNA-2), Dharmacon D-060224-03 (siRNA-3), Dharmacon D-060224-04 (siRNA-4), Invitrogen 4390771-s90877 (siRNA-5), Invitrogen 4390771-s90878 (siRNA-6), Dharmacon smart pool L-060224-01 (siRNA-7) and Origene 866091-SR46066C (siRNA-8). For mouse PLXDC2, siRNAs tested included Dharmacon D-059538-01 (siRNA-1), Dharmacon D-059538-02 (siRNA-2), Dharmacon D-059538-03 (siRNA-3), Dharmacon D-059538-04 (siRNA-4), Invitrogen 4390771-n380220 (siRNA-5), Invitrogen 4390771-n380229 (siRNA-6), Dharmacon smart pool L-059538-01 (siRNA-7), and Origene 866094-SR416812C (siRNA-8). For mouse LRP-1, siRNAs tested included Origene SR423695A-866095 (siRNA-1) and SR423695B-866096 (siRNA-2). Control siRNA was from Invitrogen. The most effective siRNA was transfected through reverse transfection using RNAiMAX in antibiotic free culture medium at 50 nM concentration with a cell splitting ratio of 1:5. To achieve a high transfection rate and knockdown effect, reverse transfection was performed twice consecutively following the manufacturer's protocol. At 48 hr after transfection, the cells were reverse transfected again using the same siRNA for the second round of knockdown. Functional assays were performed 48 hours after the second round of reverse transfection.

### Gene expression analysis

Gene expression levels were analyzed by RT-PCR. Briefly, total RNA was extracted from cells with a kit (Qiagen, Hilden, Germany). Total RNA concentration was measured by Nanodrop (Thermo Scientific). Total RNA was then used to generate cDNA by using ThemoScript Reverse transcriptase (Life Technologies). Mouse PLXDC1 was amplified by 5'-GGAGGCAGAAGGCAAGACATGCG-3'and 5'-CGTGGAGGCCGAGCAGTGCTGA-3'. Mouse PLXDC2 was amplified by 5'-CTGCCAGCCGGGAT CTGTGGGTTAACATAGACC-3' and 5'-GGGAAGTGGAGTCATCTCCACAGCTGAGATGTTGG-3'.

### Copurification studies

Rim-tagged proteins were purified using the anti-Rim antibody- conjugated to CNBr-activated Sepharose 4 Fast Flow beads (Amersham/GE Healthcare). Briefly, cells were washed once with HBSS and lysed in well with 1% Triton X-100 in HBSS and protease inhibitors for 30 min on ice. Cell lysate was spun at 16,000×$g$, 4°C for 10 min to remove insoluble materials. Cell lysate was applied to anti-Rim antibody conjugated beads, and rotated for 4 hr at 4°C. The beads were washed three times

using 0.1% Triton X-100 in HBSS by spinning down at 1000×$g$ for 30 s and eluted in 0.1% Triton X-100 in 0.1 M Glycine, pH = 2.3 for 15 min at room temperature. Tris (pH 9.5) was added to 0.1 M to neutralize the elution before the samples were analyzed. HA-tagged proteins were detected using a monoclonal anti-HA antibody. To compare homooligomerization and heterooligomerization, anti-Rim purification was performed 24 hr after cells were transfected with Rim-tagged PLXDC1 (20%), HA-tagged PLXDC1 (40%) and untagged PLXDC2 (40%) in one experiment and Rim-tagged PLXDC2 (20%), HA-tagged PLXDC2 (40%) and untagged PLXDC1 (40%) in another experiment. Copurified receptors were detected either by anti-HA antibody or antibody specific to PLXDC1 or PLXDC2. Polyclonal antibodies against the N-terminal peptide of human PLXDC1 (SPQPGAGHDEGPGSGWAAKGTVRG) and the N-terminal peptide of human PLXDC2 (KPGDQILDWQYGVTQAFPHTE) were produced by conjugating the peptides to KLH before immunization of rabbits (Genemed Synthesis, San Antonio, TX). Antibodies were purified from rabbit crude sera using the corresponding peptide conjugated to Affigel (Bio-Rad, Hercules, CA).

### Real-time analysis of PEDF-mediated dissociation of receptor oligomerization by fluorescence resonance transfer (FRET)

CFP and YFP proteins were fused to the C-terminus of PLXDC1 and PLXDC2 to detect oligomerization of PEDF receptors. Three glycine linkers were added between YFP/CFP and the C-terminal tail of PLXDC1 or PLXDC2. FRET analysis was performed similarly as described (*Kawaguchi et al., 2011*). Briefly, membranes were prepared from HEK293 cells that coexpress PLXDC1-CFP and PLXDC2-YFP. CFP-YFP FRET was measured in black flat bottom 96-well plates (Microfluor 2, Thermo Scientific) using simultaneous dual emission optics in POLARstar Omega with excitation filter 422-20 and emission filters 470-12 and 530-10. The background signal of each reaction was measured before PEDF was added to the membrane suspension to initiate the reactions. The signal from each time point was the average of 20 measurements. After all the measurements were done, the signals were calculated as the ratio of emissions at 530 nm over emissions at 470 nm to observe the dynamic change in FRET. To crosslink the C-terminal free cysteine using BMOE (Pierce), membrane preparations were made in PBS and 5 mM EDTA. BMOE was added to the membrane suspension at a concentration of 2 mM. The reaction was carried out at room temperature for 1 hour. Concentrated DTT solution was added to 5 mM to quench the reaction. After incubation at room temperature for 10 min, 1 ml of HBSS/HEPES (HBSS with 10 mM HEPES, pH 7.5) was added to the membrane suspension. After the membranes were pelleted down, the resulting membrane pellets were washed once and resuspended in HBSS/HEPES for FRET measurement.

## Acknowledgements

Supported by the Early Career Scientist Award of Howard Hughes Medical Institute (HS) and UCLA Claude Pepper Older Americans Independence Center (HS). We thank Drs Ernest Wright, Dean Bok, Ken Philipson, Gabriel Travis, Xian-Jie Yang, Jeremy Nathans and Lily Wu for helpful discussion and/or suggestions on the manuscript.

# Additional information

### Funding

| Funder | Author |
| --- | --- |
| Howard Hughes Medical Institute (HHMI) | Hui Sun |
| Claude Pepper Older Americans Independence Center (UCLA) | Hui Sun |

The funders had no role in study design, data collection and interpretation, or the decision to submit the work for publication.

### Author contributions

GC, MZ, RK, HS, Designed the experiments, contributed to the 7-year discovery phase of this project, contributed to the characterization and mechanistic study of the receptors, wrote the paper; MK, Contributed to the 7-year discovery phase of this project, contributed to the characterization and

mechanistic study of the receptors; MA-U, Contributed the cone-derived cells; JD, MT-S, Contributed to the 7-year discovery phase of this project

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
