## [Decision Letter]

Thank you for sending your work entitled “Identification of PLXDC1 and PLXDC2 as the Transmembrane Receptors for the Multifunctional Factor PEDF” for consideration at *eLife*. Your article has been favorably evaluated by Charles Sawyers (Senior editor) and 2 reviewers, one of whom, Michael Brown, served as Reviewing editor.

The Reviewing editor and the other reviewer discussed their comments before we reached this decision, and the Reviewing editor has assembled the following comments to help you prepare a revised submission.

Reviewer 1 raised the following major issue: The current paper presents the discovery of a pair of cell surface receptors for PEDF, a peptide that is reported to exert a bewildering array of different actions on various cells and pathologic processes. The authors provide evidence for a novel mechanism of action whereby the two receptors (PLXDC1 and PLXDC2) are present in an inactive heterodimer in the basal state. PEDF binds to either PLXDC1 or PLXDC2, which causes the heterodimer to dissociate, thereby allowing each of the receptors to transmit regulatory signals.

The authors use a variety of cell types and a variety of cell biological methods to establish the mode of action of PEDF. In general, the experiments are well performed. The binding experiments involve overexpression, which always has the potential to create artificial interactions. However, the RNAi-mediated knockdown experiments support the model.

A concern regarding the proposed dimer dissociation mechanism is that it requires that each cell type express stoichiometrically equal amounts of both receptors. If one of the isoforms is expressed in excess it should be free to signal in the absence of PEDF. This conclusion seems inconsistent with the results of Figure 2 in which overexpression of PLXDC1 does not increase IL-10 secretion in RAW 267.4 cells in the absence of added PEDF. A similar discrepancy occurs in the experiment of Figure 4 in which knockdown of neither PLXDC1 nor PLXDC2 causes the death of SVEC4-10 cells in the absence of added PEDF. The same can be said of Figure 5 in which knockdown of neither PLXDC1 nor 2 produces the neurotrophic effect in the absence of added PEDF. Clearly, PEDF must be doing something other than or in addition to dissociating PLXDC heterodimers.

In the Discussion the authors cite several publications that apparently show that various cells express relatively high amounts of either PLXDC1 or PLXDC2, apparently in the absence of overexpression of the other isoform. Again the dimer dissociation model would suggest that these cells should express the relevant function of the overexpressed PLXDC isoform without requiring PEDF. To deal with this issue in the current paper the authors should report the relative amount of each isoform in each of the cell types under study.

Reviewer 2 raised the following major issue: First, the case made for LRP1 as a co-receptor for PLXDC1 and 2 is solely based on data shown in Figure 3 and that evidence is weak. Essential controls are missing. For instance, the LDL receptor, which has the same general structure as the LRP1 extracellular domains 1 to 4, could have been used as a negative control. Various cell lines expressing or lacking LDLR or LRP1 exist and could have been used as straightforward controls in binding and co-IP experiments. Moreover, ALL the LRP1 domain constructs used in Figure 3 seem to bind to either PLXDC1, 2 or both. The difference in intensity of the pulled down proteins can be largely attributed to the variable expression level of the LRP1 domains. Thus, there seems to be no discrimination. It seems odd that PLXDCs would bind to at least 4 different sites on the receptor. Again, a negative control using e.g. the LDLR would help distinguish non-specific from specific interactions.

Lastly, the siRNA result shown in Figure 3 is not convincing either. Is that difference between control siRNA and LRP1 siRNA really convincing? I presume siRNA-2 was used to knock down LRP1. Finally, the interpretation of the data hinges on IL-10 secretion. However, the figure shows that IL-10 secretion is already increased in the absence of PEDF. Thus, LRP1 may control IL-10 secretion independent of PEDF. Panel B thus does not add direct support to the hypothesis that LRP1 interacts with PLXDCs.

Fortunately, the primary impact or validity of the study does not depend on an interaction of LRP1 with PLXDCs. I thus suggest to either remove Figure 3 entirely, or alternatively add substantially more data to support the claim.

---

## [Author Response]

*[…] A concern regarding the proposed dimer dissociation mechanism is that it requires that each cell type express stoichiometrically equal amounts of both receptors. If one of the isoforms is expressed in excess it should be free to signal in the absence of PEDF. This conclusion seems inconsistent with the results of*
Figure 2
*in which overexpression of PLXDC1 does not increase IL-10 secretion in RAW 267.4 cells in the absence of added PEDF. A similar discrepancy occurs in the experiment of*
Figure 4
*in which knockdown of neither PLXDC1 nor PLXDC2 causes the death of SVEC4-10 cells in the absence of added PEDF. The same can be said of*
Figure 5
*in which knockdown of neither PLXDC1 nor 2 produces the neurotrophic effect in the absence of added PEDF. Clearly, PEDF must be doing something other than or in addition to dissociating PLXDC heterodimers*.

*In the Discussion the authors cite several publications that apparently show that various cells express relatively high amounts of either PLXDC1 or PLXDC2, apparently in the absence of overexpression of the other isoform. Again the dimer dissociation model would suggest that these cells should express the relevant function of the overexpressed PLXDC isoform without requiring PEDF. To deal with this issue in the current paper the authors should report the relative amount of each isoform in each of the cell types under study*.

The model we proposed focuses on homooligomers. We proposed that PEDF dissociates receptor homooligomers (e.g., PLXDC1 dimers) to activate the receptor. Because our model involves homooligomers, not heterooligomers, the model is not affected by their stoichiometry. We did not study or discuss heterooligomers in the original manuscript. However, whether these receptors preferentially form homooligomer or heterooligomer is an interesting question. We have performed a new experiment to answer this question and found that both PLXDC1 and PLXDC2 prefer to form homooligomers (Figure 5—figure supplement 1). This result is consistent with the homooligomer model we proposed in the original manuscript. We agree with Reviewer 1’s comment that a model that depends on heterooligomer formation is problematic due to the requirement of strict stoichiometry in the expression of both receptors in all cell types.

*[T]he case made for LRP1 as a co-receptor for PLXDC1 and 2 is solely based on data shown in*
Figure 3
*and that evidence is weak. Essential controls are missing. For instance, the LDL receptor, which has the same general structure as the LRP1 extracellular domains 1 to 4, could have been used as a negative control. Various cell lines expressing or lacking LDLR or LRP1 exist and could have been used as straightforward controls in binding and co-IP experiments. Moreover, ALL the LRP1 domain constructs used in*
Figure 3
*seem to bind to either PLXDC1, 2 or both. The difference in intensity of the pulled down proteins can be largely attributed to the variable expression level of the LRP1 domains. Thus, there seems to be no discrimination. It seems odd that PLXDCs would bind to at least 4 different sites on the receptor. Again, a negative control using e.g. the LDLR would help distinguish non-specific from specific interactions*.

*Lastly, the siRNA result shown in*
Figure 3
*is not convincing either. Is that difference between control siRNA and LRP1 siRNA really convincing? I presume siRNA-2 was used to knock down LRP1. Finally, the interpretation of the data hinges on IL-10 secretion. However, the figure shows that IL-10 secretion is already increased in the absence of PEDF. Thus, LRP1 may control IL-10 secretion independent of PEDF. Panel B thus does not add direct support to the hypothesis that LRP1 interacts with PLXDCs*.

*Fortunately, the primary impact or validity of the study does not depend on an interaction of LRP1 with PLXDCs. I thus suggest to either remove*
Figure 3
*entirely, or alternatively add substantially more data to support the claim*.

As suggested, we have deleted Figure 3 since this figure is not a central point of this paper. We originally included these experiments because a previous study identified PLXDC1 as a membrane protein downregulated by LRP-1 (Gaultier et al, 2010). However, due to the extremely large size of LRP-1, detailed mapping of its interaction with PLXDC1 and PLXDC2 is a separate topic that demands systematic work in the future. siRNA-2 for LRP-1 in the Figure reproducibly knocked down LRP-1 expression by more than 80%. We also tried PEA-10 cells (Lrp-1 +/-) and PEA-13 cells (Lrp-1 -/-), but encountered the difficulty that currently no PEDF-related assay has been established in these cellular models.